# Identification and Quantification of Pteridines in the Wild Type and the *ambar* Mutant of *Orius laevigatus* (Hemiptera: Anthocoridae)

**DOI:** 10.3390/insects16080756

**Published:** 2025-07-23

**Authors:** Yolanda Bel, Amador Rodríguez-Gómez, Pablo Bielza, Juan Ferré

**Affiliations:** 1Laboratory of Biotechnological Control of Pests, Institute of Biotechnology and Biomedicine (BIOTECMED), Universitat de València, Burjassot, 46100 València, Spain; yolanda.bel@uv.es; 2Departament of Genetics, Universitat de València, Burjassot, 46100 València, Spain; 3Biocontrol Selection Laboratory, Departamento de Ingeniería Agronómica, Universidad Politécnica de Cartagena, 30203 Cartagena, Spain; amador.rodriguez@upct.es (A.R.-G.); pablo.bielza@upct.es (P.B.)

**Keywords:** *Orius* nymphs, erythropterin, leucopterin, 7-methylxanthopterin, xanthopterin, isoxanthopterin, pterin, biopterin, XDH, thrips predator, insect pigments

## Abstract

*Orius laevigatus* (Fieber) is a predator, particularly of thrips, widely used in the biological control of agricultural pests. Nymphs are yellowish in all their developmental stages. In 2021, an orange-colored strain was established from a wild nymph, and it was determined that the coloration was due to an autosomal recessive mutation, *ambar*. With the aim of determining the reason for the different color in the *ambar* mutant nymphs, insect pigments belonging to the pteridine family (fluorescent compounds that work as pigments in many insect species) were studied and identified using different approaches. This is the first time that these compounds have been identified in this species. The results showed that the orange pigment erythropterin accumulated at significantly higher levels in the *ambar* mutant, indicating that the orange coloration is a consequence of its accumulation.

## 1. Introduction

*Orius laevigatus* (Fieber, 1860) (Hemiptera: Anthocoridae) is an important predator which is widely used to control thrips pests in biological control programs [1]. This insect species is being mass-produced by numerous biocontrol companies to be released mainly in vegetable crops in greenhouses. Interestingly, *O. laevigatus* can also induce defensive responses in plants, such as sweet peppers, which can help repel pests and attract natural enemies [2]. Recent work has demonstrated that artificial selection can enhance key traits in *O. laevigatus*, including pesticide resistance [3,4], improved performance on sub-optimal diets [5], and increased body size linked to greater fecundity and predation capacity [6,7]. Collectively, these advances further consolidate *O. laevigatus* as a versatile and increasingly efficient tool for integrated pest-management programs.

So far, no mutations have ever been reported for this insect species, except for the *ambar* mutation in 2022 [8]. This mutation, autosomic and recessive, confers an orange color to the nymphs, clearly distinguishable from the wild-type color which is yellow (Figure 1). Mutations affecting external body coloration may become very useful markers in studies of dispersion, adaptation, and population genetics, and can also serve as indicators for studies on mating and sexual competition.

Although several pigments can contribute to the external color of insects, pteridines are, along with melanin and ommochromes, the most important [9]. Pteridines are a family of aromatic bicyclic compounds, present in all living species. In addition to their function as nitrogen excretory substances and as essential cofactors of many metabolic reactions, they can also function as pigments, as visual filters, and for external signaling in insects. The most commonly found pteridine pigments in Hemiptera are erythropterin (orange), xanthopterin (yellow), and 7-methylxanthopterin (yellow) (a compilation of pteridines found in Hemiptera can be found in Table S4 of the Suppl. Meth. of [10]). Nevertheless, to date there is no study addressing the identification of the pteridine pigments in *O. laevigatus* or any other species of the *Orius* genus. In the present work, we aimed at identifying the pteridines in this species and determining the differences responsible for the color in the *ambar* mutant.

## 2. Materials and Methods

### 2.1. Source of Insects and Pteridine Standards

Wild-type and *ambar* mutant colonies were maintained in the Biocontrol Selection Lab at the Universidad Politécnica de Cartagena, Spain. They were reared under controlled conditions [8]. The *ambar* colony originated from a mutant individual that appeared spontaneously from the wild-type population as described elsewhere [8].

Pterin, biopterin, leucopterin, xanthopterin, and isoxanthopterin were obtained from Sigma Chem. Co. (St. Louis, MO, USA). Neopterin was a gift from Dr. W. Pfleiderer (University of Konstanz, Konstanz, Germany).

### 2.2. Separation of Fluorescent Compounds by Thin-Layer Chromatography (TLC)

Separation of fluorescent compounds from the nymphs by cellulose TLC was performed according to Ferré et al. [11]. Plates of 20 × 20 cm (Supelco HX20548816, St. Louis, MO, USA) were used. Nymphs were homogenized manually in microtubes with pestle in MAW (methanol-glacial acetic acid-water, 4:1:5, by vol.) under dim light. The homogenates were centrifuged at 21,130× *g* for 5 min. Under dim red light, the supernatants were spotted at a corner of TLC plates in aliquots of 3 µL followed by air drying until all the volume of the sample was delivered. The spotted samples were subjected to two-dimensional chromatography in the dark. Isopropanol—2% ammonium acetate (1:1 by vol.) was used for the first dimension for 8 h. After drying overnight, the plates were subjected to the second dimension for 1 h 45 min using 3% ammonium chloride. Fluorescent spots were visualized after two to four days under long-wave UV light (366 nm).

### 2.3. Partial Purification of ambar Nymph Extracts by Column Chromatography

Separation of UV-absorbing compounds from *ambar* nymph extracts was performed by size exclusion chromatography (SEC) in a Sephacryl HiPrep 26/60 column (320 mL of bed volume) equilibrated in distilled water. Two milliliters of an extract of around 500 nymphs in 5 mL of MAW were injected into the column at a flow rate of 1.5 mL/min. The column was eluted with distilled water to separate the blue-fluorescent compounds and then the solvent was changed to 0.1 M NaCl to elute the orange-fluorescent compounds that were strongly retained in the column. The absorbance was followed at 360 nm and 455 nm. Fractions of 10 mL were collected.

### 2.4. Identification of Fluorescent Compounds by LC/MS/MS

Identification of the cellulose-extracted compounds was carried out using a TripleTOF™ 6600+ LC/MS/MS system and an Exion (AB SCIEX, Redwood, CA, USA) instrument. Chromatographic separation was carried out on a Waters UPLC C_18_ column 1.7 μm (2.1 × 50 mm) Acquity UPLC BEH.C18 from Waters (Cerdanyola del Vallès, Spain). The mobile phase consisted of a gradient of 14 mM ammonium formate, pH 6.8 (solvent A), and acetonitrile (solvent B). The flow rate and injection volume were 0.3 mL/min and 3 μL, respectively. The mass range of MS acquisition was 100–800 *m*/*z*. The MS is processed by an Information Dependent Acquisition (IDA) with the survey scan type (TOF-MS) and the dependent scan type (Product Ion) at 30 V of collision energy. The MS conditions were ion spray voltage of 5500 V, declustering potential of 90 V, temperature at 600 °C with curtain gas of 40 psi, ion source gas 1 at 60 psi, and ion source gas 2 at 60 psi. IDA MS/MS was performed as follows: ions that exceeded 100 CPS, ion tolerance 50 mDa, collision energy fixed at 25 V, and dynamic background subtraction activated.

### 2.5. Quantification of Fluorescent Compounds

For quantitative analysis of the fluorescent compounds, from each strain, 70 nymphs were homogenized in 175 µL of MAW. Following centrifugation, 50 µL of each sample was applied to the plates. After the two-dimensional chromatography, plates were allowed to air dry for five days. Then, fluorescent spots were visualized under UV light and the cellulose containing them was scraped off with a microspatula. The isolated compounds were extracted from the cellulose by adding 150 µL of MAW, mixing by vortexing, and letting it stand overnight. The next day, the samples were centrifuged at 21,130× *g* for 5 min, and 100 µL were transferred to 96-well plates. Fluorescence was measured from the plates using a BioTek Synergy H1 microplate reader (Winooski, VT, USA) with the wavelengths set at or close to the excitation/emission maxima of each compound. For erythropterin and xanthopterin, the wavelengths were those of the maxima in the fluorescence spectra of erythropterin (455/535 nm). Isoxanthopterin was measured at 345/410 nm. Although the fluorescence maxima for the blue-fluorescent compounds pterin and biopterin were 355/440, the wavelengths chosen for quantification were 370/445 to minimize the contribution of fluorescence from the cellulose. A blank control was prepared by scraping a similar area of the cellulose plate where no fluorescent compounds were observed and the value obtained at the different wavelengths was subtracted from those of the spots. The analyses were replicated 6 to 7 times for each strain.

### 2.6. Analysis of Xanthine Dehydrogenase (XDH) Activity

The XDH activity was measured as the conversion of pterin to isoxanthopterin. For the analysis, 19 mg of nymphs of each strain were homogenized in 350 µL of 0.1 M phosphate buffer, pH 7.5 at 4 °C, in a mortar. The homogenates were centrifuged at 21,000× *g* for 10 min at 4 °C. Small molecules were eliminated by passing the supernatant through 40 K Zeba spin desalting columns (0.5 mL bed volume) (Thermo Fisher Scientific, Waltham, MA, USA) equilibrated in the same buffer. The eluate containing the proteins was kept at 4 °C until used within one hour. The total protein content was analyzed by Bradford [12]. The enzyme reaction contained 200 µM pterin, 2 mM NAD^+^, and 80 µL of sample in a final volume of 100 µL. The enzyme activity was measured following the appearance of isoxanthopterin by measuring the fluorescence in a BioTek Synergy H1 microplate reader with the excitation wavelength of 334 nm and the emission wavelength of 412 nm (excitation/emission chosen to maximize the detection of isoxanthopterin and minimize that of pterin). The reaction was monitored every 2 min at 20 °C. The final reading was taken at 14 min, before linearity was lost. Three biological replicates were conducted. Negative controls were also set in each replicate with either without substrate or without cofactor; in both cases, there was no increase in fluorescence.

## 3. Results

### 3.1. Identification of Fluorescent Compounds Separated by TLC

Two-dimensional TLC revealed the presence of several fluorescent components in the nymphs of *O. laevigatus* (Figure 2). The most intense spots in the *ambar* mutant were spot O, an orange pigment with orange fluorescence; spot P (colorless with pale blue fluorescence); spot V (colorless with violet fluorescence); spots Y1 and Y2 (greenish-yellow fluorescence); and spots B1 and B2 (colorless with blue fluorescence).

According to the fluorescence color and *R*_f_ values (corresponding to the relative migration) from the literature and standards, spot O was tentatively identified as erythropterin, spot P as leucopterin, spot Y1 as xanthopterin, spot Y2 as 7-methylxanthopterin, spot V as isoxanthopterin, spot B1 as pterin, and spot B2 as biopterin [11,13]. Tandem mass spectrometry after separation by liquid chromatography (LC/MS/MS) of the extracted compounds from the cellulose plate confirmed their identity. The molecular mass and the fragmentation of the cellulose-extracted compounds perfectly matched those predicted by their molecular formula (Figure 3).

### 3.2. Identification of Fluorescent Compounds Separated by SEC

Column chromatography of the MAW extract of *ambar* nymphs rendered six well-separated peaks absorbing at 360 nm (the maximum wavelength of the pteridine ring) (Figure 4). The UV detector was also set at 455 nm to detect yellow compounds.

Fractions collected from each peak were concentrated by lyophilization with further solubilization in MAW. To further purify the components of each fraction, aliquots were subjected to TLC in the two solvents (Figure A1). The main fluorescent components revealed under UV light were scraped from the cellulose plate, dissolved in MAW, and analyzed by LC/MS/MS. From the fluorescence color and the LC/MS/MS identification, it was concluded that peak 1 was a mixture of biopterin, pterin, and 7-methylxanthopterin; peak 2 contained a mixture of isoxanthopterin (the main component) and xanthopterin; peak 3 only contained leucopterin; and peak 6 contained erythropterin as the main component. Peaks 4 and 5 did not show any fluorescent compound that could be visualized.

### 3.3. Quantification of Fluorescent Compounds

The most striking difference observed when comparing the TLC chromatograms of the wild-type and *ambar* strains is that, whereas the *ambar* mutant had a considerable amount of erythropterin, the wild type had almost non-measurable amounts (Figure 5). Also, leucopterin was much more intense in the wild type. Another qualitative difference between the wild type and the *ambar* mutant was that pterin increased in the latter.

The main fluorescent spots in the TLC chromatograms of the two strains were quantified by measuring the fluorescence intensity of the individual compounds isolated from the TLC plates. The fluorescent spot of 7-methylxanthopterin was difficult to precisely draw its perimeter, resulting in limited reliability of its quantification, and it was not included in the quantitative analyses. Also, attempts to quantify leucopterin were unsuccessful due to the interference of erythropterin and the background fluorescence of the cellulose. The quantitative results from six independent replicates from each insect sample are shown in Figure 6. Erythropoietin and pterin were confirmed to be significantly different in the two strains, whereas xanthopterin, isoxanthopterin, and biopterin did not show significant differences.

### 3.4. XDH Activity

Based on the proposed general pteridine biosynthetic pathway in Hemiptera (Figure 7), the accumulation of pterin and erythropterin in the *ambar* mutant could be a consequence of a deficiency in XDH activity.

To test this hypothesis, the XDH activity was measured in the extracts of wild-type and *ambar* nymphs. The results showed no differences in activity amongst both samples (Figure 8), indicating that the XDH activity was not the reason for the *ambar* phenotype.

## 4. Discussion

The analysis of pteridines in the genus *Orius* has been undertaken for the first time in this work, triggered by the interest in knowing the reason for the change in body color of the nymphs and adults of a spontaneous mutant, *ambar*. TLC was applied to separate, in two dimensions, the fluorescent compounds of *O. laevigatus.* Two-dimensional TLC provides, in addition to the *R*_f_ in two solvent systems, the color of the separated compounds at visible and under UV light, which strongly contributes to their identification.

Since most water-soluble fluorescent compounds in insects are pteridines [14], identification of the TLC-separated fluorescent compounds was addressed by comparison with pteridine synthetic standards and by LC/MS/MS analysis after separation by either TLC or LC/TLC. The combined approaches identified erythropterin, leucopterin, xanthopterin, 7-methylxanthopterin, isoxanthopterin, pterin, and biopterin in *O. laevigatus*, with erythropterin, isoxanthopterin, and pterin as the most abundant in the *ambar* mutant.

Qualitative comparison of the TLC chromatograms of the wild type and the *ambar* mutant revealed three clear differences: erythropterin was in trace amounts in the wild type whereas leucopterin was increased, and the spot corresponding to pterin was much fainter than in the chromatogram of the *ambar* mutant. These differences were further confirmed (except for leucopterin) by quantitative analysis of the fluorescence intensity, as shown in Figure 6. Erythropterin, in addition to its fluorescence under UV light, absorbs light in the visible spectrum, being an orange-red pigment. Therefore, its accumulation in the *ambar* mutant seems to be the reason for the orange color of nymphs. Indeed, the correlation of erythropterin content and insect body pigmentation has been described in other Hemiptera such as *Dysdercus* spp. [13] or in *Pyrrhocoris apterus* after the analyses of erythropterin in several body color mutants [15].

The pteridines identified in this study had previously been reported in Hemiptera, though never together in the same species (as a review, see Table S4 of the Suppl. Meth. of [10]). Our study, with the identification of seven pteridines, contributes to a better understanding of the biosynthesis of these compounds in Hemiptera. According to a holistic proposed pathway of pteridines in insects [14] and in Hemiptera [10], their biosynthesis in *O. laevigatus* would most likely be as shown in Figure 7. Biopterin, a colorless compound, appears in the TLC chromatogram most likely as an oxidation product of tetrahydrobiopterin, an essential cofactor for the biosynthesis of aromatic amino acids and biogenic amines [16,17]. From the precursor 6,7-dihydropterin, the pathway gives rise to two branches, one that produces the colorless compounds pterin and isoxanthopterin, and the other to the yellow-to-red pigments xanthopterin, erythropterin, and 7-methylxanthopterin, the latter thought to be a spontaneous degradation product of erythropterin [18]. Although we have detected 7-methylxanthopterin in fraction 1 of the SEC separation of pteridines from the *ambar* mutant, its detection in TLC plates was uncertain and for this reason, it was not included in the quantitative analysis.

According to the biosynthetic pathway and taking into account the fact that the *ambar* mutation is recessive, i.e., loss of function [8], a possible hypothesis is that the *ambar* gene controls one step in the pathway. A gain of function, such as an increase in a step of the synthesis of erythropoietin, is very unlikely, the mutation being recessive. Considering the increase in the amount of erythropterin and pterin in the *ambar* mutant, and the reduced level of leucopterin compared to the wild type, a possible explanation could be that the *ambar* gene controls the step catalyzed by XDH. A reduced activity of this enzyme would make the levels of pterin increase, and xanthopterin would preferentially be converted to erythropterin instead of leucopterin. However, our results do not support this hypothesis unless a lack of a specific XDH activity related only to pteridine biosynthesis in specific organs could explain the mutant phenotype. An alternative explanation would be that the *ambar* mutant has blocked a step downstream of erythropterin, causing this compound to accumulate when it is otherwise converted to other compounds (such as di-pteridines, like pterorhodin [15], or other non-colored derivatives) in the wild type. Because of the lack of information on the possible processing of erythropterin [19,20,21], this hypothesis is difficult to test.

## 5. Conclusions

The present work characterizes the changes in the pteridines responsible for the orange color of nymphs in the *ambar* mutant of *O. laevigatus*, a valuable marker for biological and ecological studies of this important thrips predator widely used as a biological control agent. The holistic approach identified for the first time the most abundant pteridines in nymphs, both in the wild type and in the *ambar* mutant: erythropterin, leucopterin, 7-methylxanthopterin, xanthopterin, isoxanthopterin, pterin, and biopterin. The quantification of the differences between the two strains showed that the color in the *ambar* nymphs is due to the accumulation of the orange pigment erythropterin.

## Figures and Tables

**Figure 1 insects-16-00756-f001:**
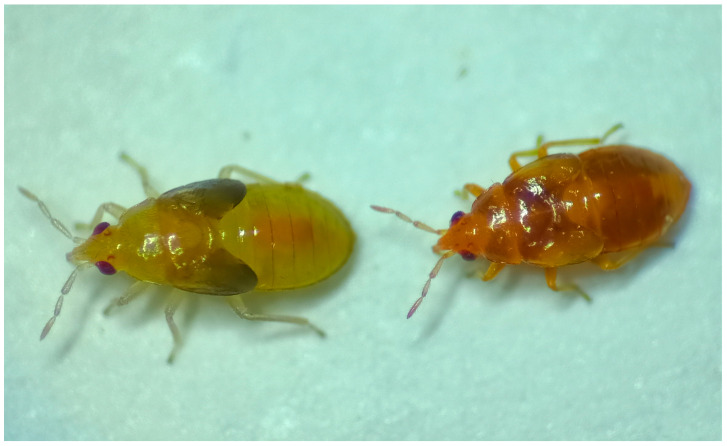
Nymphs of the wild type (**left**) and *ambar* mutant (**right**) of *O. laevigatus*.

**Figure 2 insects-16-00756-f002:**
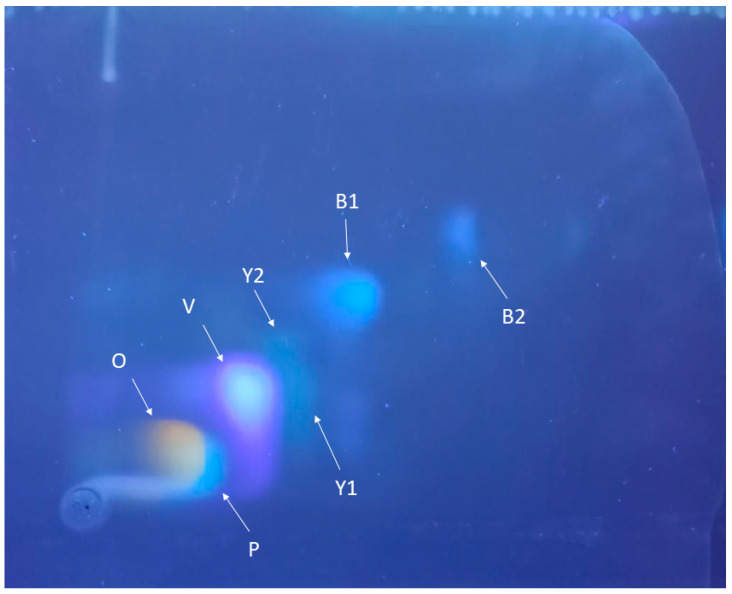
Separation of *O. laevigatus* fluorescent compounds of the *ambar* mutant by two-dimensional TLC.

**Figure 3 insects-16-00756-f003:**
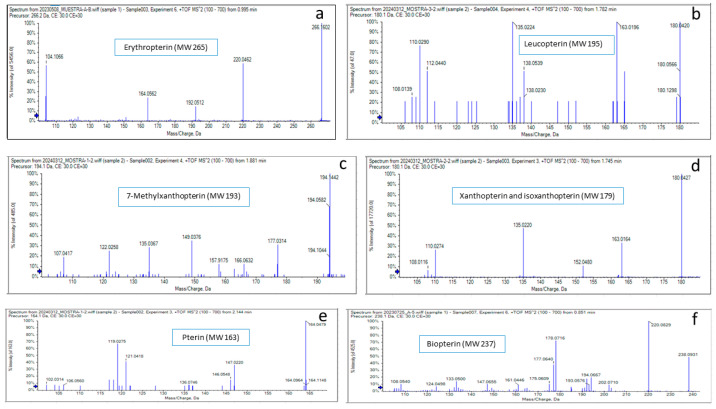
Identification of pteridines in *O. laevigatus* using LC/MS/MS. (**a**) Erythropterin, (**b**) leucopterin, (**c**) 7-methylxanthopterin, (**d**) xanthopterin and isoxanthopterin, (**e**) pterin, and (**f**) biopterin.

**Figure 4 insects-16-00756-f004:**
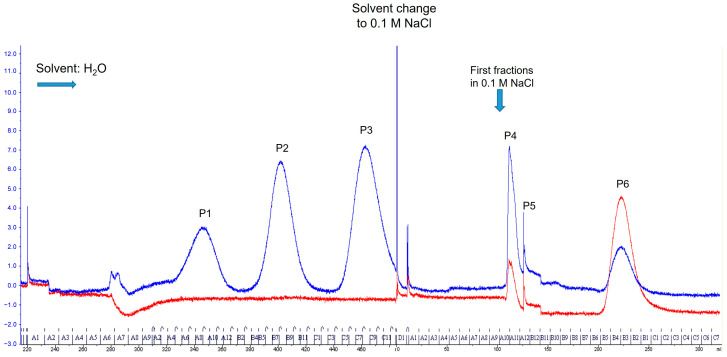
Separation of *O. laevigatus* fluorescent compounds by SEC. Red line: Absorbance at 455 nm. Blue line: Absorbance at 360 nm.

**Figure 5 insects-16-00756-f005:**
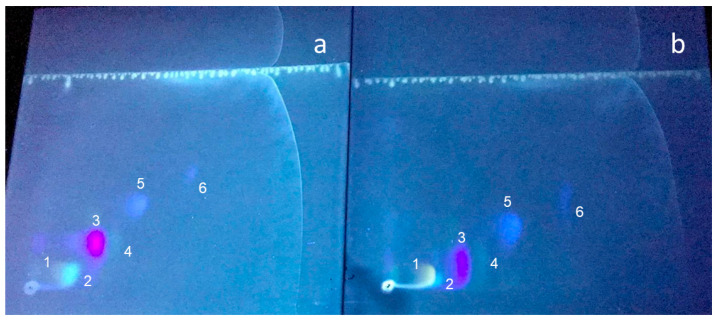
TLC chromatograms of extracts of (**a**) the wild type and (**b**) the *ambar* mutant. 1, Erythopterin; 2, leucopterin; 3, isoxanthopterin; 4, xanthopterin; 5, pterin; 6, biopterin.

**Figure 6 insects-16-00756-f006:**
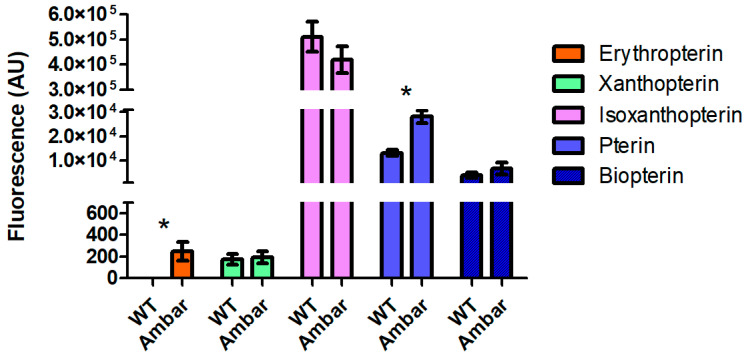
Fluorescence intensity (in arbitrary fluorescence units) of the fluorescent compounds in the wild type and *ambar* mutant of *O. laevigatus*. Bars represent the mean of 4 to 6 replicates. Asterisks indicate statistically significant differences (*p* < 0.05) as determined by *t*-test.

**Figure 7 insects-16-00756-f007:**
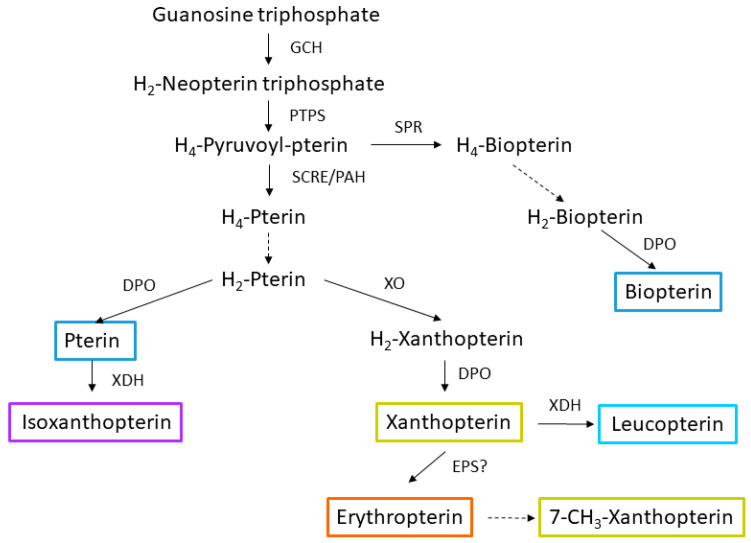
Proposed pteridine pathway in Hemiptera [14]. GCH: GTP cyclohydrolase; PTPS: Pyruvoyl-tetrahydropterin synthase; PAH: Phenylanaline hydroxylase; DFR: Dihydrofolate reductase; SPR: Sepiapterin reductase; SCRE: Side chain releasing enzyme; DHPO: Dyhydropterin oxidase; XDH: Xanthine dehydrogenase; EPS: Erythropterin synthase. Dashed line, spontaneous? Pterins detected in *Orius laevigatus* are boxed with their respective fluorescence color.

**Figure 8 insects-16-00756-f008:**
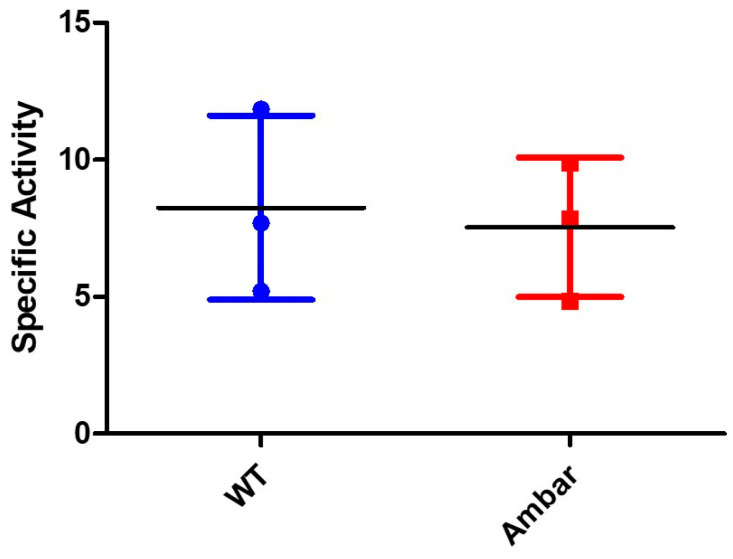
XDH-specific activity (units represent the increase in fluorescence per min and mg of protein) in extracts of nymphs from the two strains. The graph represents the means and standard deviations of three replicates.

## Data Availability

The original contributions presented in this study are included in the article. Further inquiries can be directed to the corresponding author.

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
