# Peer review of "Identification and Quantification of Pteridines in the Wild Type and the ambar Mutant of Orius laevigatus (Hemiptera: Anthocoridae)"

_insects, 2025, doi:10.3390/insects16080756_

Round 1
Reviewer 1 Report
Comments and Suggestions for Authors
I find this manuscript interesting and informative. However, I have a major complaint about section 2, Materials and Methods, where the source of tested insects is not mentioned anywhere in the paper, especially in the M&M section.
Also, a few methodological details were not explained precisely.
All comments are in an uploaded Word file.

Author Response
Reviewer comments insects-3674523
Line 45 I would suggest to add the year of describing species, it is usually added next to the authors name; but it just a suggestion
The year has been added
Section 2. Materials and Methods: I don’t see if you mention anywhere in the MS the source of insects. You mention the source of pteridine and used methods but not insects. Were they are collected from the field or reared in laboratory? How can you be sure there are not two species e.g. Also, do you have any data about adults?
We have now added the following information in the section 2.1:
Wild type and ambar mutant colonies were maintained in the Biocontrol Selection Lab at the Universidad Politécnica de Cartagena, Spain. They were reared under controlled conditions. The ambar colony was originated from a mutant individual that appeared spontaneously from the wild type population as described elsewhere [8].
As described in reference 8, adults do not present appreciable differences with the naked eye.
Line 95/section 2.3 You don’t mention any reference in this section. Why exactly you used 535 nymphs for extraction
There is no reference of the methodology employed because we just applied SEC to the separation of molecules by their size and affinity for the colum matrix.
We wrote 535 nymphs because they were all that we had at that moment. Since this is not critical for the purification, we have now changed it to “around 500 nymphs”.
Section 2.4 Again, there is no reference cited, did you use this methodology for the first time, or you/someone else used similar so that the modification can be mentioned?
The identification of organic molecules is routinely used by the LC/MS/MS facility of our university. And yes, it was applied to the pteridines by the first time. This is why we do not have a reference for it.
Line 116 Why exactly you used 70 nymphs?
Preliminary trials indicated that this proportion of nymphs/MAW volume gave the appropriate fluorescence intensity in TLC.
Line 192 delete second was that is not necessary
Done
Line 210 You are mentioning “..general pteridine biosynthetic pathway in Hemiptera”, how can you claim it is the same for all Hemiptera, shouldn't you add some reference for this?
We have now added a recent review on the biosynthesis of pteridines in the legend of the figure which recompiles all information on pteridines in Hemiptera, including the thorough study of Vargas-Lowman et al. (ref. 10).
Line 247 when first mention, species name must be full, not shorten. For Pyrrhocoris apterus (L.)
Done
Reviewer 2 Report
Comments and Suggestions for Authors
Dear authors
The work is interesting and well structured, but I suggest being more specific in the presentation of the results (see suggestion inside the document).

Author Response
Line 40. Why, is keyword “Biological Control”. were predation tested?
Response: We have deleted this keyword since the work does not deal with this aspect
Line 116. 70 wild type nymphs and 70 ambar mutant ???? make that clear
Response: We have changed the sentence making it clearer.
Line 168. Fig. 3 have 6 figures (a-f). please describe each one.
Response: letters (a-f) have been added to the figure panels.
Legend to Fig. 3. Are 6 figures. write the legend for each. please.
Response: The 6 different panels have been explained in the legend.
Line 190. Two figure (left and right) a,b describe each; adjust the legend too
Response: We have added a and b in the figure and left and right has been substituted by a and b in the legend.

Round 2
Reviewer 1 Report
Comments and Suggestions for Authors
Section 2. Materials and Methods:
Line 77: Now you mentioned the source of insects, but if you say they were reared under controlled conditions, you should specify those conditions. Or, if that is the standard for your lab, e.g., then cite the work where it is well explained.
Author Response
The reference of the original paper where the rearing methodology is fully described is N0. 8, which was at the end of the paragraph. Now, to make it clearer, we have included this referenc after the sentence of rearing conditions.